# Curcumin Scaffold as a Multifunctional Tool for Alzheimer’s Disease Research

**DOI:** 10.3390/molecules27123879

**Published:** 2022-06-17

**Authors:** Haijun Yang, Fantian Zeng, Yunchun Luo, Chao Zheng, Chongzhao Ran, Jian Yang

**Affiliations:** 1Shanghai Engineering Research Center of Organ Repair, School of Medicine, Shanghai University, Shanghai 200444, China; yhj10222021@163.com (H.Y.); yunchunluo1109@163.com (Y.L.); 2School of Public Health, Xiamen University, Xiamen 361000, China; zftcpu@163.com; 3PET Center, School of Medicine, Yale University, New Haven, CT 06520, USA; c.zheng@yale.edu; 4Athinoula A. Martinos Center for Biomedical Imaging, Massachusetts General Hospital and Harvard Medical School, Boston, MA 02129, USA

**Keywords:** Alzheimer’s disease, amyloid-β, tau protein, curcumin scaffold, AD diagnosis

## Abstract

Alzheimer’s disease (AD) is one of the most common neurodegenerative disorders, which is caused by multi-factors and characterized by two histopathological hallmarks: amyloid-β (Aβ) plaques and neurofibrillary tangles of Tau proteins. Thus, researchers have been devoting tremendous efforts to developing and designing new molecules for the early diagnosis of AD and curative purposes. Curcumin and its scaffold have fluorescent and photochemical properties. Mounting evidence showed that curcumin scaffold had neuroprotective effects on AD such as anti-amyloidogenic, anti-inflammatory, anti-oxidative and metal chelating. In this review, we summarized different curcumin derivatives and analyzed the in vitro and in vivo results in order to exhibit the applications in AD diagnosis, therapeutic monitoring and therapy. The analysis results showed that, although curcumin and its analogues have some disadvantages such as short wavelength and low bioavailability, these shortcomings can be conquered by modifying the structures. Curcumin scaffold still has the potential to be a multifunctional tool for AD research, including AD diagnosis and therapy.

## 1. Introduction

Alzheimer’s disease (AD) is a multi-faceted neurodegenerative disease in the elderly population with complicated pathogenesis. According to the World Alzheimer Report in 2018, the global number of dementia patients is currently up to 50 million people, of which 50–75% are AD patients. The number will reach 152 million by 2050 and that means, every three seconds, one person is to become AD in the world [1]. For the elderly, AD has become the fourth largest killer following heart disease, cancer and stroke [2]. However, there are no treatments that can delay or prevent the disease’s progression. As a direct consequence, AD is becoming a huge social and economic burden to the world. In recent years, the activity and mechanism between protein and drug binding were exploited, such as microtube affinity-regulating kinase (MARK4) and AChE inhibitors, which were beneficial to the AD therapy aid [3,4,5].

Generally speaking, there are two remarkable neuro-histological hallmarks in postmortem AD patient brain tissues: extracellular amyloid-β (Aβ) plaques and intracellular neurofibrillary tangles of hyperphosphorylated tau protein [6]. It has been proposed that the abnormality of Aβ appears far earlier than tau abnormality, and the Aβ abnormality can induce hyperphosphorylation of tau protein and neuroinflammation [7,8,9]. Therefore, the amyloid cascade theory is considered to be related to the pathology of AD. Based on the research guidelines published by NIA-AA in 2018, Aβ abnormality has been considered to be an early biomarker for diagnosis in the early stages of AD.

According to amyloid cascade theory, Aβ is produced by cleavages of amyloid-β precursor protein (APP) by β- and γ-secretase and is precluded by α-secretase activity. While Aβ42 and Aβ40 are the two major Aβ species, Aβ42 is more aggregation-prone and readily forms neurotoxic oligomers and is more prevalent than Aβ40 in plaques [10,11]. Normally, the N-terminal segment has the capacity to bind metal ions, such as Fe^2+^, Cu^2+^ and Zn^2+^, which is simultaneously accompanied by the production of reactive oxygen species (ROS) [12]. During the Aβ aggregation, there are several Aβ sub-species including monomers, dimers, oligomers, and fibrils/aggregates. During the aggregation process, all the Aβ sub-species involved in the process are equilibrated [13]. In this equilibration process, once Aβ monomers reach a critical high concentration, over-accumulation of Aβ begins. Aβ monomers can quickly gather into large molecular weight polymeric precipitation, which turns into oligomers and eventually fibrils/aggregates [14,15,16,17,18] that will produce toxic effects on the surrounding neurons and synapses, and cause synaptic membrane damage, resulting in neuronal cell death [14,15,19,20,21]. The amyloid cascade hypothesis is shown in Figure 1.

Tau protein can stabilize the internal skeleton of nerve cells in the brain. Studies show that the total amount of tau proteins in AD brains is increased significantly, and the increased tau proteins are usually in the form of excessive abnormal phosphorylation, which means that if the tau protein is abnormally phosphorylated, it is easy to form paired helical filaments (PHF), which can lead to neurofibrillary tangles (NFTs). NFTs are highly related to the occurrence of AD [22,23,24]. Clearly, developing new imaging probes that target NFTs is becoming of great significance for the accurate diagnosis of AD.

In recent years, neuroinflammation is considered a highly relevant biomarker in AD, which is closely associated with oxidative stress (OS). In fact, oxidative stress always plays an important role in AD pathogenesis [25,26]. Mounting research evidence revealed that the concentration of reactive oxygen species (ROS) in AD brains is much higher than in that of healthy brains [27,28,29]. ROS, including superoxide radical, hydroxyl radical and hydrogen peroxide, always contribute to the pathophysiology of neurodegenerative disorders. Although ROS can modulate cell survival, once the balance of ROS was broken, it will be accompanied by oxidative damage. Multiple sources contributed to ROS production, such as mitochondria dysfunction, reactions involving peroxisomal oxidases, NAD(P)H oxidases, cytochrome P-450 enzymes, over-accumulation of metal ions and aggregation of Aβs or Tau tangles [27,30,31,32]. Recently, the relationship between OS and AD has been revealed, and the research shift is from fundamental research to clinical treatment, such as the use of antioxidants (vitamin E/C, curcumin, GSH) [33,34] and chelating agents.

It is well known that many countries have the habit of eating curry, especially in India, and curry contains abundant curcumin which has preventive and therapeutic effects on AD. That may be one reason for the low incidence of dementia in India. This phenomenon was drawing researchers’ attention in the AD field. Curcumin is a natural product of polyphenols with a diketone structure, the trans double bond and the keto-enol tautomerism make it have a long-conjugated carbon chain, which can result in several different fluorescent properties and functions. Remarkably, the neuroprotective effects of curcumin scaffold on AD have been extensively studied, such as the inhibition of Aβ accumulation, prevention of tau hyperphosphorylation and aggregation, as well as having anti-inflammatory, antioxidant, and metal ions-complexing properties [35].

Though curcumin and its analogues have some disadvantages, such as short wavelength, low bioavailability, poor water solubility, poor stability in solution, and the characteristics of intestinal first-pass and hepatic metabolism, these shortcomings could be conquered by modifying the structures. They still have the potential to be imaging agents or therapeutic agents in AD research. Numerous curcumin analogues have been used as a valuable molecular scaffold to decipher the fundamental pathologies of AD. Some of them have been developed to be new near-infrared fluorescent probes or PET tracers for AD diagnosis; some of the analogues are exploited to be curative molecules for AD therapy or therapeutic monitoring. Herein, we summarized the progress of curcumin scaffolds in the AD research field over the past year.

## 2. Curcumin Scaffold as a Tool for Understanding AD

Aβ plaques represent a classical pathology which is found in the AD brain. Aβ peptide usually undergoes conformation changes from monomer to oligomers and fibers that are rich in β-sheets [36]. Martin et al. analyzed the dynamics and energetics between curcumin and 24-peptide in Aβ fibrils using molecular dynamics simulations. The results show that particular hydrophobic sites existed on the protofibril surface that can bind with curcumin, and this binding is usually associated with the complexation of curcumin with dimers, trimers, or oligomers, even at the end of the fibrillary growth axis. The hydrophobic interaction and solvation effects may contribute to the good binding between curcumin and protofibril. These interactions may be the key to reducing the toxicity of Aβ oligomers and changing the Aβ accumulation pathway to form non-toxic aggregates [37]. Zhao et al. demonstrated that curcumin is also a potential Aβ-toxic inhibitor because it has a high tendency to interact with certain Aβ residues which were highly likely to form hydrogen bonds with curcumin. Additionally, curcumin was observed to act as a β-sheet breaker when it was inserted into Aβ protein. The π-π stacking interactions frequently existed between the aromatic ring of curcumin and proteins such as histidine, tyrosine, and phenylalanine [38]. Although the interactions are temporary, they indirectly contribute to decreasing β-sheet content [39]. Gestwicki et al. found that hydroxyl substitution in aromatic rings on both sides of curcumin was essential for inhibiting Aβ by structure–activity relationship, and the length and flexibility of the linker between aromatic rings should be kept within an appropriate range [40]. Curcumin has ketone-enol tautomerism and exists in the form of enol in solution. Studies have shown that the form of enol exhibits anti-Aβ aggregation properties, not the β-dione form [41].

Besides inhibiting Aβ aggregation, some researches showed that curcumin has the capability to inhibit hyperphosphorylated tau protein aggregation. Molecular docking studies predicted that curcumin had a possible binding site in the tau microtubule region. A strong hydrophilic interaction between curcumin and tau D225 residues was revealed by Panda. Furthermore, the electrostatic interaction might contribute to blocking the tau-tau interaction, which was considered to be the most crucial factor in the process of tau aggregation [42].

Recently, oxidative stress is deemed to be another risk factor in AD [28]. Reactive oxygen spices (ROS) can interact with lipids, proteins, and nucleic acids, which can cause irreversible damage to neurons in the brain [43]. The diketone and phenolic groups in the curcumin structure can prevent the production of a variety of reactive oxygen species. In addition, the β-dione group and the hydroxyl group can complex with metal ions, such as copper ions, ferrous ions, and zinc ions [44]. Whereas ions are necessary for the production of hydroxyl radicals, so curcumin exerts antioxidant activity directly or indirectly.

It is well known that curcumin has fluorescence properties, which is attributed to the unique structure of the diphenylenone, and the isomerization transition between the ketone form and the enol form also gives curcumin many unique photochemical properties. Therefore, curcumin can be used as sensitive material for the detection of chemical substances. Curcumin can form complexes with Cu^2+^, Fe^2+^, Zn^2+^ and other cations through the structure of 1,3-diketone that can carry out keto-enol isomerization. This complexation could lead to the increase of solubility in water and the change of color [45]. Importantly, curcumin can cross the blood-brain barrier, so it might be used as a candidate in brain studies. In addition, curcumin can specifically bind to Aβ aggregates and abnormal tau proteins, which has been reported for staining experiments, suggesting that curcumin can bind to specific groups in proteins and produce changes in optical properties [46]. Therefore, curcumin has the potential to be developed as an imaging probe candidate for AD diagnosis.

Due to a variety of effects on AD, curcumin and its analogues have the potential to be a drug for AD therapy and can act as a chemical fluorescence probe for AD with great research value.

## 3. Curcumin Scaffold as a Tool for AD Diagnosis

In recent years, curcumin and its scaffold has been utilized as fluorescence probes for animal studies and preclinical research in AD [47]. Lots of probes including near-infrared fluorescent probes and PET agents were designed and synthesized, which have good biological properties such as long wavelength, good stability in serum, good BBB penetration and so on. Imaging agents designed from the curcumin scaffold for AD diagnosis are shown in Table 1.

### 3.1. Fluorescent Probe for Detecting Insoluble Aβ

Extracellular Aβ senile plaques play a critical role in AD onset according to the amyloid cascade hypothesis. Despite the relationship between AD severity and Aβ accumulation plaques not being linear, the monitoring of Aβ is still a useful tool for predicting AD development and providing a precise index for Aβ-related treatments. Recent reports revealed that curcumin and its analogues have the potential for Aβ detection [66]. In 2011, Zhang et al. reported that a curcumin analogue named BMAOI-14 [48] displayed good binding affinity for Aβ42 fibrils (*K*_d_ = 830 nM) and low binding affinity (*K*_d_ = 71 μM) for bovine serum albumin (BSA). This chemical probe had the capability to bind Aβ plaques with high specificity in both APP mouse and AD patients’ brain tissues. However, the short emission wavelength and rapid clearance in the brain and large molecular weight limited it for in vivo study. Similarly, Yanagisawa et al. studied FMeC1 [49], another curcumin analogue, which can bind with Aβ aggregates; however, its short emission wavelength hindered its application scope. Ran et al. devoted efforts to developing some curcumin-based Aβ fluorogenic probes. In the first attempt, they designed a near-infrared fluorescent probe termed CRANAD-2 (Figure 2A), a curcumin derivative with 2,2-difluoro-1,3,2-dioxaborines as an acceptor and two *N*,*N*′-dimethyl groups as donor groups in 2009 [50]. When bound to Aβ plaques, CRANAD-2 displayed excellent optical characteristics, with a 715 nm emission wavelength, a reasonable log p-value, as well as great sensitivity (70-fold) and good binding affinity (*K*_d_ = 38.69 nM). CRANAD-2 can discern the difference between 19-month-old APP/PS1 mice and WT mice (Figure 2B,C). Nevertheless, CRANAD-2 is incapable of detecting soluble Aβ (monomers and oligomers) and has a sluggish washout rate in the brain (2 min/30 min = 2.4). In 2020, Wu et al. reported a new NIRF called CAQ [51] that is based on CRANAD-2 using a quinoline functional group to replace an aromatic ring (Figure 2A), while the Curcumin difluoroborate moiety was preserved in the structure. In CAQ, one side of 4-(dimethylamino) donor group was kept, which could lead to intramolecular rotation effects and lower the background fluorescence signal. A quinoline group was applied to improve BBB penetration and biocompatibility due to its excellent physicochemical and photophysical properties. CAQ exhibits an emission wavelength of 725 nm (DMSO) and relatively good affinity (*K*_d_ = 78.89 nM). After i.v. injection for 30 min, the fluorescence intensity was significantly higher in 10-month-old 5XFAD mice than that in age-matched control mice, indicating that probe CAQ was capable of imaging Aβ aggregates (Figure 2D,E).

### 3.2. Fluorescent Probe for Detecting Soluble Aβ

Mounting evidence has revealed that the burden of Aβ deposits does not have good correlations with cognitive impairment, and researchers have turned their attention to the soluble Aβ forms, such as oligomers. In particular, small soluble Aβ oligomers could inhibit hippocampal long-term potentiation during the in vitro and in vivo study, which can induce synapse degeneration in AD brains [67]. Nevertheless, multiple factors render it challenging to develop novel NIR fluorescent probes to detect oligomers with superior selectivity. For instance, the various Aβs (monomers, oligomers, and aggregates) almost have the same peptide unit and they are devoid of rational designing principles. Moreover, it is hard to dynamically monitor oligomers in a real-time manner in vivo due to their transitory and heterogeneous properties.

Lots of efforts have been made to exploit new probes for Aβ oligomers with high specificity. The Aβ13–20 fragment (HHQKLVFF) with hydrophilic/hydrophobic regions and structural stereo-hindrance compatibility plays an important role during Aβ aggregation. The strategy for matching this fragment could enhance affinity to soluble Aβ peptides. As previously mentioned, CRANAD-2 has no capability for detecting soluble Aβ species because its symmetric structure does not match the Aβ13–20 fragment with hydrophobic and hydrophilic properties. Therefore, Ran et al. developed an asymmetric CRANAD-2 analogue, the structures are shown in Figure 3.

An aniline and a more hydrophilic pyridyl moiety were introduced in CRANAD-58 [52]. In vitro study demonstrated that it has good fluorescence qualities (ex = 630 nm, em = 750 nm), with a fluorescence intensity increase of 91.9- and 113.6-fold for Aβ40/Aβ42 monomers respectively. However, compared to CRANAD-2, its binding affinity decreased (Aβ40: *K*_d_ = 105.8 nM, Aβ42: *K*_d_ = 45.8 nM). In order to increase binding ability, CRANAD-3 was designed by Ran and co-workers though replacing phenyl rings with pyridyl to introduce possible hydrogen bonds [53]. CRANAD-3 displayed high interaction with Aβ40/42 monomers, dimers, and oligomers (*K*_d_ = 24 ± 5.7 nM, 23 ± 1.6 nM, 16 ± 6.7 nM, and 27 ± 15.8 nM, respectively). CRANAD-3 (Figure 4A,D) and CRANAD-58 (Figure 4B,E) can both distinguish between 4-month-old APP/PS1 and WT mice. In 2017, Ran et al. made a significant breakthrough in developing a NIR probe that is selective to soluble Aβ species. The stereo-hindrance of CRANAD-3 was adjusted by adding phenoxy-alkyl chains to the dioxaborine’s central carbon. CRANAD-102 [54], a hit as a result of this strategy, showed substantial selectivity for soluble Aβ over insoluble Aβ with strong affinity (7.5 and 275 nM, respectively). In addition, CRANAD-102 could dynamically monitor the concentration of soluble Aβ in 5-month-old and 12-month-old AD transgenic mice (Figure 4C,F). In 2022, Liu group developed nine fluorescent probes to detect both soluble and insoluble Aβ species in APP/PS1 mice based on the structure of CRANAD-58. Probe 9 showed the best BBB penetration and could selectively map the Aβ plaques in both brain parenchyma and cerebral angiopathic areas [55]. Interestingly, Ran et al. discovered some of the curcumin analogues such as CRANAD-X (X = 2, 3, 30, 58, 61 and 102) could be utilized to differentiate transgenic AD mice and WT mice through NIRF ocular imaging (NIRFOI) [56]. The result showed that the huge detection margin by NIRFOI may be a high-efficiency diagnosis method in AD research due to the eye’s minimal opacity.

However, CRANAD-102 is unable to distinguish Aβ oligomer from monomeric Aβs. To overcome this obstacle, Yang et al. found that there is a triangular cavity in the structure of three types of oligomers such as trimers, hexamers and dodecamers. Furthermore, an interesting phenomenon revealed that Phe19 and Val36 residues are retained and exposed to solvent in the structure of Aβ oligomers. [57,68]. This discovery provided the possibility to develop a new probe with high specificity to oligomers. Thus, they proposed that an excellent probe for Aβ oligomers can insert into the protein cavity to increase binding affinity, and it may form π–π stacking interactions and hydrophobic contacts with Phe19/Val36 residues to improve specificity. Based on these rationales, PTO-29 was designed with a “V-shaped wedge” conformation by direct connection between the benzene and dioxa-difluoroborinine rings, which could retain steric hindrance while also strengthening the conjugate effect to achieve a longer emission (Figure 5A). The *K*_d_ values for oligomers and plaques of PTO-29 were 24.8 nM and 2.7 µM, and the selectivity is 10.9-folds, suggesting a much higher affinity for oligomers. However, the brain washout rate was slow due to its higher lipophilicity. In order to change the hydrophilicity, the hydroxyethyl group was incorporated into the design so as to increase the affinity toward the Aβ oligomers, optimize the lipophilicity, and improve in vivo pharmacokinetics. PTO-41 [58] displayed an emission wavelength maximum at 695 nm in PBS (Figure 5A). Furthermore, PTO-41 displayed greater brain uptake and faster elimination than PTO-29 (Figure 5B). More importantly, PTO-41 was successfully used to detect Aβ oligomers and distinguish APP/PS1 mice from age-matched wild-type mice in vivo study using an animal imaging system. These favorable findings were ascribed to the appropriate emission and good BBB penetration. The ability to target Aβ oligomers, which were generated at the early phase of AD, means that PTO-41 could be useful in early AD diagnosis compared with Aβ plaque-targeting probes.

These curcumin analogues can detect Aβs in vitro and in vivo; however, the probes always have low quantum yield (QY). Ran et al. substituted the phenyl rings with pyrazoles to improve the brightness of these probes in order to overcome their low QY limitation [59,69]. CRANAD-28 has a high QY (Φ = 32%) in PBS as expected. The result showed that CRANAD-28 is a good two-photon imaging probe which can detect Aβ plaques and cerebral amyloid angiopathies (CAAs) (Figure 6A), but it is not suitable for in vivo NIRF imaging due to its short emission wavelengths (578 nm). In addition, Yang et al. reported AD-1 by incorporating 2-aminoethyl diphenyl borate (DPBA) as a stabilizer into the curcumin scaffold to improve stability and BBB penetration ability [60]. AD-1 has an emission wavelength of 704 nm and can monitor all Aβ forms. Furthermore, it was successfully used to image 4-month-old APP/PS1 mice in vivo studies (Figure 6B,C).

### 3.3. Fluorescent Probe for Detecting Tau Protein

Hyperphosphorylated microtubule-associated tau (MAPT) protein deposits are another classical hallmark in Alzheimer’s patients. Considering the intimate relationship between tau aggregation and neuronal injury, a probe which has the ability to detect tau deposits may be beneficial for predicting the progression and state of AD. The first curcumin analogue for detecting tau aggregates was described by Chong et al. in 2015. Probe 1c (Figure 7) has good fluorescence characteristics (*K*_d_ = 0.77 µM) and can monitor tau aggregates in SHSY-5Y cells that have been transfected with tau-GFP [70]. In 2017, Chong et al. disclosed a series of fluorescent probes consisting of a length-extendable-bridge between a substituted difluoroboron-diketonate with an N,N-dimethylaniline moiety [71]. Probe 2e (Figure 7) has outstanding fluorescent qualities (em = 690 nm) and displays high specificities to tau protein compared with Aβ and BSA (*K*_d_ = 8.8 and 6.2) [70]. Furthermore, in a human AD brain tissue sample, probe 2e can almost completely overlap with the region of tau aggregates antibody labeling.

### 3.4. Fluorescent Probe for Detecting ROS

Besides amyloid-β and tau proteins, numerous studies on AD have focused on the reactive oxygen species (ROS). Oxidative stress has been assumed to play a key role in the etiology of AD. The cytoskeletal alteration in neurons, which was associated with irreversible cellular dysfunction, can ultimately lead to neuronal apoptosis. This is the most important component of cellular oxidative damage in AD. Abundant ROS could be generated during the process of Aβ aggregation. Some chemical reactions, such as the Fenton reaction, are always accompanied by Aβ crosslinking. The inflammatory cytokines such as TNF will accumulate due to the excessive ROS, which could attract microglia and cause numerous ROS surrounding the plaques, contributing to neuronal loss [72,73]. Therefore, detecting ROS in the AD brain is advantageous.

In 2016, Ran et al. first designed a curcumin-based probe, named CRANAD-88 [61], which has the capability to cascade signal amplification for detecting H_2_O_2_ in AD brains (Figure 8A,C). In the presence of H_2_O_2_, CRANAD-88 displayed amplified NIR fluorescence signals at three levels. After H_2_O_2_ treatment, the boronated moiety in CRANAD-88 could be removed, and the fluorescence emission peak at 690 nm was redshifted to 730 nm. Moreover, CRANAD-88 exhibits a fast and obvious response toward H_2_O_2_ and shows higher selectivity compared with other ROS species (•OH, O_2_•^−^, TPHBP, OCl^−^, and NO•). The in vivo result showed that the fluorescent signal was higher in APP/PS1 mice than in WT mice. This verified for the first time that monitoring the changes of H_2_O_2_ in AD mice before and after treatment with H_2_O_2_ scavenger is feasible (Figure 8C,D). CRANAD-61, another NIRF probe designed by the same group in 2017 [62], was used to detect ROS at the micro- and macro-levels in different aged APP/PS1 mice (Figure 8B). In the structure of CRANAD-61, the oxalate moiety has the function of reacting with ROS, resulting in a wavelength shift. A significant blueshift (from 810 to 570 nm) was observed after incubation with H_2_O_2_ and this shift could be applied for the two-photon and NIR imaging. The spectral change was used as a radiometric readout in two-photon imaging, which can detect the local concentration of ROS around plaques. When CRANAD-61 was exposed to ROS, it could turn into another probe: CRANAD-5 [62]. The plaques in two-photon imaging showed different rates of conversion between the two probes through the signal changes. If the radiometric measurement reveals a high conversion rate, it means there is a high concentration of ROS at these locations. Previous research showed plaques with “hotspots” contain more Aβ42 and are more toxic to the nervous system [25]. This result aligns well with the reports. In 18-month-old mice, the in vivo experiment revealed that the NIRF signal was significantly higher in the control group than in the transgenic group (Figure 8E,F). This is due to the fact that there are higher ROS levels in the AD brain compared to a healthy brain. Moreover, CRANAD-61 also showed the capability to monitor the age-related increase in ROS levels in AD mice. Thus, CRANAD-61 could be a candidate agent to monitor ROS changes in various AD pathological conditions or therapy programs.

### 3.5. PET Tracers

Positron emission tomography (PET) imaging is a high-performance tool widely applied in clinical studies due to its excellent sensitivity, unlimited tissue depth penetration, and safety. So far, four PET tracers had been approved by the FDA for use in clinical study including Florbetapir, Florbetaben, Flutemetamol and Flortaucipir.

Recently, curcumin analogues were exploited as PET tracers. Ryu et al. designed eight novel curcumin derivatives [63]. Among these agents, [^18^F]8 displayed the highest binding affinity (*K*_i_ = 0.07 nM) (Figure 9), and it was successfully radiolabeled and tested as a possible Aβ plaque PET tracer. In normal mice, partition coefficient testing and biodistribution revealed that [^18^F] 8 had an acceptable lipophilicity and initial brain uptake. [^18^F] 8 has the property of metabolic stability in the brain, according to metabolism studies. These findings imply that [^18^F] 8 is a potential radioligand for imaging Aβ plaques. Rokka et al. utilized a facile one-pot method to synthesize [^18^F]4 under nucleophilic 18F-fluorination conditions [64]. In vitro studies demonstrated that [^18^F] 4 had a high binding affinity with Aβ plaque in the transgenic APP mouse. In vivo and ex vivo studies revealed that this PET tracer has fast clearance from the blood and moderate metabolism. Unfortunately, it showed lower BBB penetration. Yang et al. investigated the potential of hemicurcuminoids which could detect all Aβ species as the PET imaging ligand in 2019 [65]. Using the dioxaborine scaffold as a foundation, they created F-CRANAD-101 (Figure 9), a hemicurcuminoid with a pyrazole group to adjust the hydrophobicity and a cyclobutyl moiety to reduce the reactivity of the α-carbonyl site under the base condition. A tosylate group was allowed for an SN_2_ reaction with ^18^F, enabling the achievement of a PET imaging tracer. F-CRANAD-101 had a high affinity for all Aβ species. Despite its limited fluorogenic properties, F-CRANAD-101 still had the capability to image Aβ plaques and cerebral amyloid angiopathy (CAA) through two-photon microscopy. The PET imaging results showed that the signals are much higher in 5 and 14-month-old APP/PS1 AD mice. As the second generation PET tracer, [^18^F]-CRANAD-101 exhibited the potential capability of monitoring the abnormal aggregation of Aβ in early diagnosis [65].

### 3.6. NIRF Probe for Monitoring Therapy

As we know, AD is an incurable disease. Although there are some drugs approved by the FDA, these test drugs did not show obvious effectiveness. Therefore, it is desirable for clinical research to develop effective therapeutics and imaging probes which have the capability to assist drug discovery. Recently, PET tracers such as ^11^C-PiB and ^18^F-AV-45 have been successfully applied in evaluating the efficacy of test AD drugs in clinical studies. Nevertheless, there are still lots of disadvantages of using PET tracers, such as the high cost of tracers, radioactivity, and insensitivity to Aβ species, particularly for soluble forms, which hindered the usefulness to monitor drug therapy, especially in the small animals. NIRF imaging is a cheap, acceptable depth penetration and noninvasive operation technology suitable for small animal studies. Therefore, NIRF imaging could have the potential for monitoring AD therapy in animal studies.

In 2015, Ran group reported a curcumin analogue, named CRANAD-3 [53], which was capable of monitoring therapeutic effects after drug treatments. After the APP/PS1 mice were treated with BACE-1 inhibitor LY2811376, which can lower soluble Aβ species and lead to a 60% decrease in the soluble Aβs in the cortex region [74], the fluorescence signal of CRANAD-3 was 33% lower than the signal from the same mice before treatment (Figure 10A,B). Additionally, 4-month-old APP/PS1 mice were treated with CRANAD-17 to exploit CRANAD-3′s capacity for detecting therapeutic effects. CRANAD-17 has been reported to reduce copper-induced Aβ cross-linking in in vitro study. The imaging signals from CRANAD-3 demonstrated that the group treated with CRANAD-17 showed lower NIRF signals than the control group. The reduction value reached approximately 25% (Figure 10C,D). The ELISA analysis result showed a 36% decrease in Aβ40 in brain extracts (Figure 10E), and the plaque counting experiment demonstrated a 56% reduction following CRANAD-17 treatment compared to the control group treated with Th.S (Figure 10F,G).

In 2016, Ran et al. reported a NIRF probe, named CRANAD-88, which could distinguish the APP/PS1 mice from WT mice due to its high specificity to H_2_O_2_ [61]. More importantly, it has the capability to monitor H_2_O_2_ scavenging in in vivo studies. After being treated with sodium pyruvate, which was a hydrogen peroxide (H_2_O_2_) scavenger, the difference between the fluorescent signal from APP/PS1 mice and WT mice was significantly decreased. However, the difference between APP/PS1 mice and WT was reported after one month, which means H_2_O_2_ was re-produced. In 2018, another curcumin analogue CRANAD-102 was utilized to test the feasibility of near-infrared fluorescence ocular imaging (NIRFOI) [56]. The results demonstrated that the NIRFOI signal was approximately 70% lower from APP/PS1 mice treated with LY2811376 for 3 days than that from the same mice before treatment. All the evidence revealed that NRIF with curcumin scaffold has the potential to monitor AD therapy.

## 4. Curcumin Scaffold as a Tool for AD Therapy

Recent studies have revealed the close connection between curcumin scaffold and AD. Research uncovered that curcumin can prevent Aβ aggregation and cross the blood-brain barrier (BBB), reach brain cells, repair the nerve and participate in the treatment of AD, such as ameliorating cognitive decline and improving synaptic functions in animal models [75].

### 4.1. Aβ Inhibition

In the last decade, curcumin has been considered an efficient inhibitor for β-amyloid (Aβ) aggregation. Mounting evidence has revealed that curcumin has a variety of anti-amyloid properties. It has been noted that curcumin and its derivatives are reported to have some potential binding sites for Aβ protein, to block its assembly. Some in vitro studies showed that curcumin has the capability of inhibiting the fibrillar Aβ formation from Aβ monomer and also destabilizing preformed fibrillar Aβ [47]. Meanwhile, several in vivo studies showed that curcumin could not only promote disaggregation of existing amyloid deposits but also prevent the formation of new amyloid deposits, and even reduce the size of remaining deposits [76,77]. Curcumin could also bind to Aβ oligomers and block its accumulation to lower the toxicity [78,79]. Curcumin was also found to decrease Aβ peptides levels in in vitro studies by inhibiting the expression of presenilin 1, which has a close relationship to γ secretase and leads to the generation of Aβ [80]. In 2020, Huang reported curcumin could inhibit BACE1 (β-amyloid precursor protein Cleaving Enzyme) gene expression in SH-SY5Y cells at transcriptional and translational levels. Furthermore, this result revealed that the signaling in nuclear factor kappa B is involved in the regulation between curcumin and BACE1 [81].

### 4.2. Tau Inhibition

The formation of the neurofibrillary tangles with tau hyperphosphorylation plays an important role in the pathogenesis of AD [82,83]. The prevention of hyperphosphorylation of tau has been the most significant hotspot of AD research [84]. Several studies showed that curcumin can prevent tau hyperphosphorylation and decrease neurotoxicity [85]. Curcumin could inhibit α-synuclein accumulation and increase the protein solubility, which could lead to decreasing levels of soluble tau dimers and elevated heat shock proteins contained in the process of tau clearance in AD mice [86]. HSP 70 and HSP 90, two heat shock proteins, closely enhance the solubility of tau protein and promote tau-microtubule interaction [87]. In 2017, Panda reported that curcumin has the capability to bind the adult tau and the fetal tau. The dissociation constants were 3.3 ± 0.4 µM and 8 ± 1µM, respectively. Molecular docking studies showed that curcumin had a possible binding site in the tau microtubule-binding region, which may lead to inhibiting tau aggregation and disintegrating preformed tau oligomers. More importantly, it revealed that curcumin has the capability to break the formation of β-sheets in protein structure to inhibit aggregation [42]. In the same year, Sun reported that curcumin could be utilized as the lead compound to study the correlation between tau protein and Caveolin-1 (a marker protein of membranal caveolae). After being treated with curcumin, the expression of Caveolin-1, tau protein and their relationship were detected, and the potential mechanism was also explored [88].

### 4.3. Metal Ion Binding

Metal ion dyshomeostasis has been recognized as a cofactor which might cause cellular death or severe dysfunction in neurodegenerative disorders such as Alzheimer’s disease (AD) [89,90]. The Aβ misfolding process, which is associated with aggregation, is related to the metal irons (i.e., copper, iron, and zinc) that could be found in both the core and rim of the AD plaques [91,92]. Studies have shown that the levels of metal ions (iron, zinc, and copper) in the brain of AD patients are 3–7 times higher than those in the normal brain [93]. Animal studies have confirmed that curcumin has an anti-inflammatory effect. Curcumin is a metal chelating agent, which can exhibit anti-AD effects. Reducing metal ions which can induce Aβ amyloid fibril formation is a possible mechanism [94,95]. Recent studies suggest that curcumin has the capability to bind metal irons [96,97]. The strategy of measuring the affinity between curcumin and different metal ions may verify the likelihood that curcumin could protect against AD through chelation. In 2013, Picciano et al. demonstrated that estimating Cu^2+^-curcumin binding affinities in the absence and presence of the Aβ peptide could provide evidence for this Cu^2+^ chelation role [98]. Moreover, a curcumin analogue named CRANAD-17 [52] exhibited the ability to inhibit Aβ42 cross-linking that was induced by copper. It raised the potential for CRANAD-17 to be considered a hit compound for AD therapy.

### 4.4. ROS Scavenging and Neuroprotective Effects

Mounting evidence has revealed that oxidative stress is a key role in the onset and progression of AD [99]. Besides Aβ protein and tau, chronic inflammation is considered another pathological hallmark of AD, which is characterized by the increased presence of activated microglia and astrocytes, accompanied by elevated expression of acute-phase proteins and proinflammatory cytokines [100,101]. During the aggregation of Aβs, numerous ROS are generated. The generated ROS can further induce the accumulation of inflammatory cytokines such as TNFα, which can affect the microglia to release more ROS [25,30]. Many studies have shown that curcumin has potential anti-inflammatory effects such as reducing levels of IL1β and oxidized protein, microgliosis in the cortex, Aβ peptides and plaque burden [101,102,103]. Moreover, curcumin also has the capability to protect PC1_2_ cells and normal human umbilical endothelial cells from oxidative stress caused by amyloid-β [104]. Curcumin, a PPARγ agonist, could improve mitochondrial function and reduce ROS leakage, which could indirectly reduce oxidative stress. Meanwhile, curcumin also can inhibit LPS-induced microglial activation by inhibiting NO production and reducing the release of pro-inflammatory cytokines such as IL-6, and IL-1β [105]. Most importantly, curcumin can attenuate microglial migration and trigger another microglial phenotype with anti-inflammatory and neuroprotective effects [106]. Numerous studies have revealed that curcumin exhibits good properties such as scavenging free radicals, reducing iron ions, chelating metal ions, and inhibiting lipid oxidation degradation [107]. Therefore, curcumin could be capable of protecting glial cells and reducing ROS.

## 5. Conclusions

There is plenty of evidence to support that the curcumin scaffold has neuroprotective effects in AD such as anti-amyloidogenic, anti-inflammatory, anti-oxidative and metal chelating. Furthermore, it has also played an important role in monitoring AD therapy. However, the short wavelength always hindered being a favorable fluorescence imaging tool in in vivo studies, and the low bioavailability extremely reduced the benefits of AD therapy. Lots of disadvantages are exhibited in the curcumin scaffold, including poor water solubility, poor stability in solution, and the characteristics of intestinal first-pass and hepatic metabolism [108]. The most critical limitation is bioavailability. Although there are many effective methods to improve the bioavailability of curcumin, the concentration of curcumin in the brain is still very low due to its fast metabolism in the blood. Some studies have shown that slow-release drugs using nanotechnology have higher efficacy and bioavailability [109]. However, further research should be carried out to verify the long-term use and safety of nanomaterials if curcumin was prepared as a nanoparticle delivered to the brain. In addition, although lots of studies have confirmed that curcumin has relatively low toxicity, high-dose curcumin use is not uncommon in most in vivo and clinical trials. Therefore, high doses of curcumin could have some toxic risks. Unfortunately, these problems have not been effectively addressed. In future studies, bioavailability, solubility and BBB penetration are the major crucial factors that are necessary to be considered during the design of probes or PET tracers with curcumin scaffold for diagnosis or therapy in AD research.

## Figures and Tables

**Figure 1 molecules-27-03879-f001:**
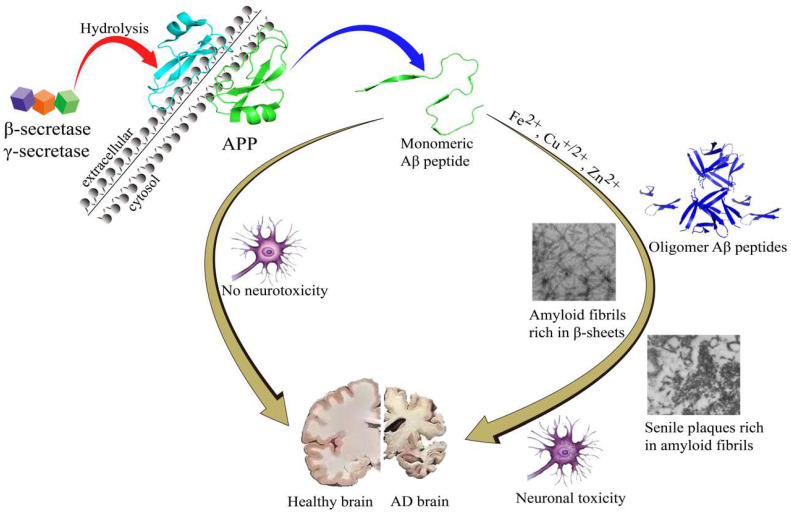
The amyloid cascade hypothesis.

**Figure 2 molecules-27-03879-f002:**
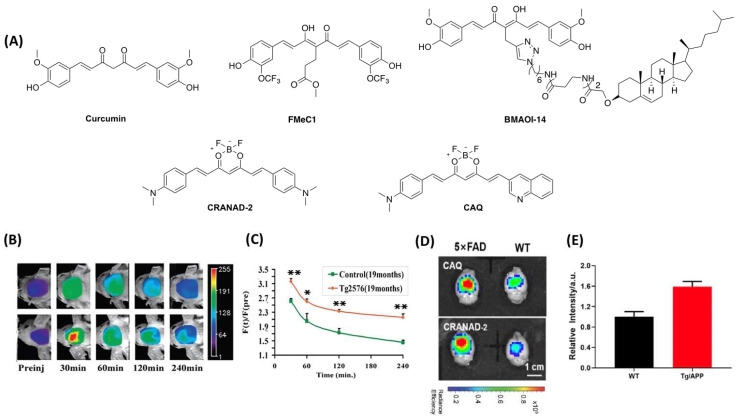
(**A**) The structures of curcumin, FMeC1, BMAOI−14, CRANAD−2 and CAQ; (**B**) Representative images of Tg2576 mice and control mice at different time points; (**C**) The fluorescence signal curves of F(t)/F(pre) between transgenic mice and the control group (* *p* < 0.05, ** *p* < 0.01); Reprinted with permission from [50]. Copyright 2009 American Chemical Society. (**D**) Representative images of 5XFAD mice and WT mice at 30 min. (**E**) Fluorescence intensity of the brain in 5XFAD and WT mice at 30 min. Reprinted with permission from [51]. Copyright 2021 American Chemical Society.

**Figure 3 molecules-27-03879-f003:**
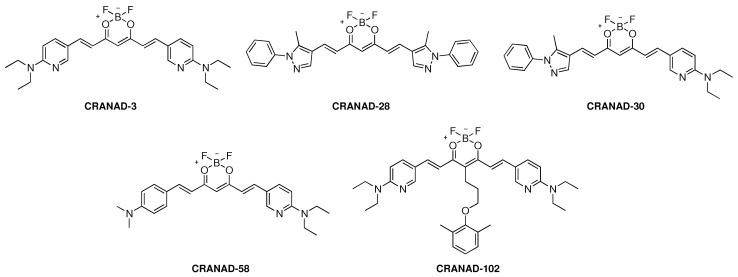
The chemical structures of CRANAD−X (X = 3, 28, 30, 58 and 102).

**Figure 4 molecules-27-03879-f004:**
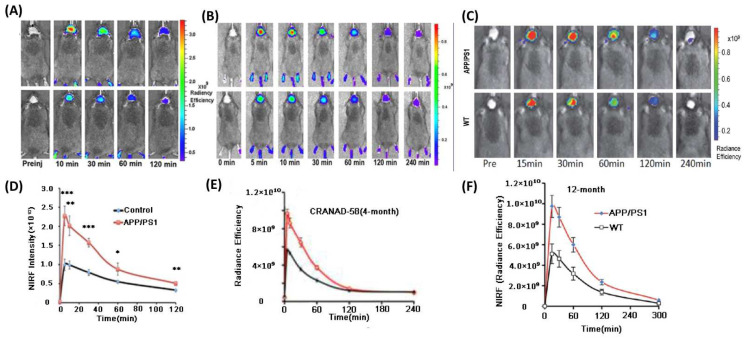
(**A**–**C**) Representative images of APP/PS1 transgenic and wild-type mice at different time points of CRANAD−3, 58 and 102, reprinted with permission from [52]. Copyright 2013 American Chemical Society. (**D**–**F**) Quantitative analysis of fluorescence signals between APP/PS1 and WT mice. * *p* < 0.05, ** *p* < 0.01, *** *p* < 0.005. Reprinted with permission from [54]. Copyright 2019 Royal Society of Chemistry.

**Figure 5 molecules-27-03879-f005:**
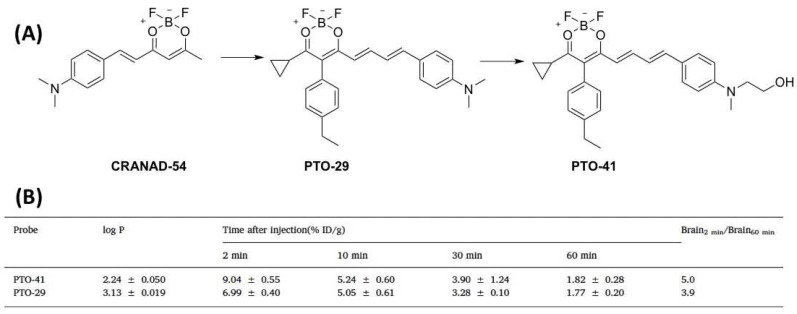
(**A**) The chemical structures of CRANAD−54, PTO−29, PTO−41 and AD−1; (**B**) Log *p* values and the brain uptake (%ID/g) of PTO−29 and PTO−41 at different time points.

**Figure 6 molecules-27-03879-f006:**
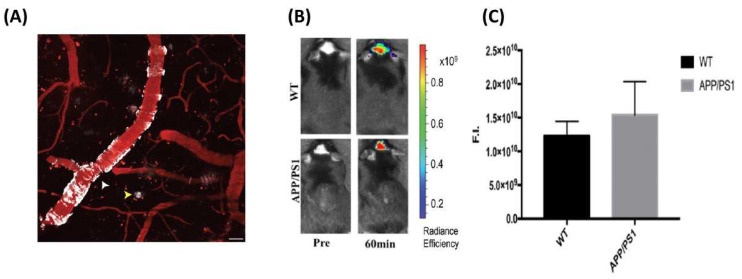
(**A**) Two-photon images of CRANAD-28 labeling in a 9-month-old APP/PS1 mouse. Both CAA and amyloid plaques were labeled (white arrow). (**B**) Representative images of APP/PS1 and WT mice after i.v. injection with AD-1 at 60 min; (**C**) Fluorescence intensity of the brain in APP/PS1 and WT mice at 60 min. Reprinted with permission from [60]. Copyright 2021 American Chemical Society.

**Figure 7 molecules-27-03879-f007:**
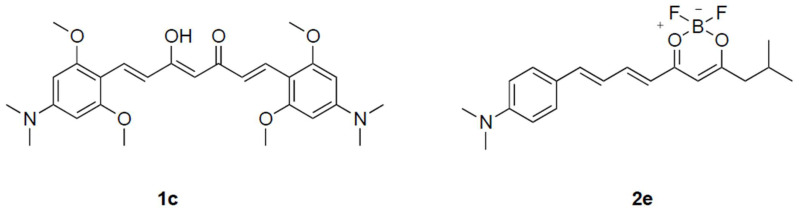
The chemical structures of probes (1c and 2e) which can detect Tau protein.

**Figure 8 molecules-27-03879-f008:**
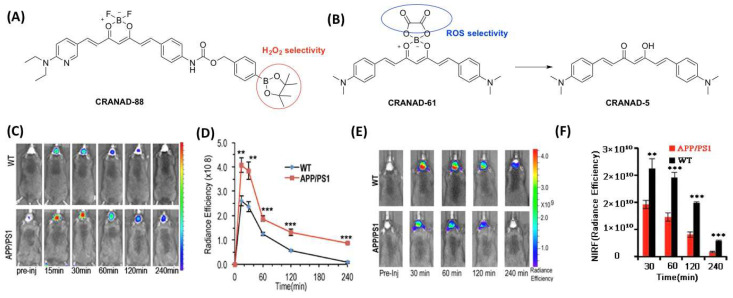
(**A**) The chemical structure of CRANAD-88; (**B**) The reaction of CRANAD-61 with ROS; (**C**,**E**) Representative images of APP/PS1 and WT mice after i.v. injection at different time points, reprinted with permission from [61]. Copyright 2016 Springer. (**D**,**F**) Fluorescence intensity of CRANAD-88 and 61 between APP/PS1 and WT mice in brain area. ** *p* < 0.01, *** *p* < 0.005. Reprinted with permission from [62]. Copyright 2017 PNAS.

**Figure 9 molecules-27-03879-f009:**
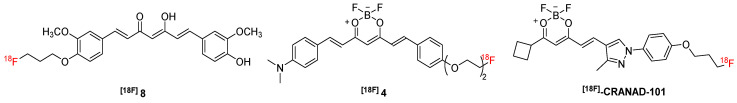
The chemical structures of different PET tracers.

**Figure 10 molecules-27-03879-f010:**
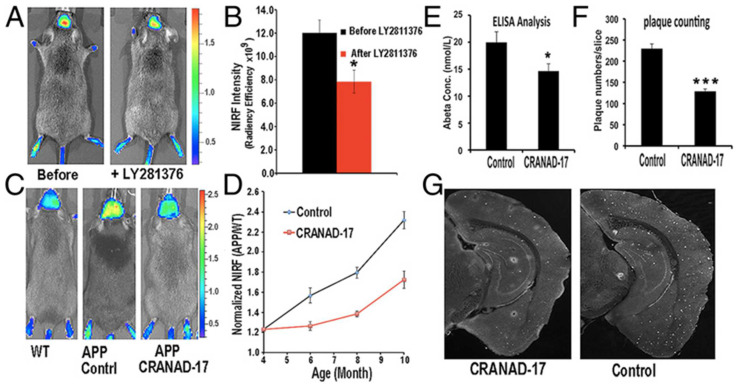
CRANAD-3 for detecting therapeutic effects with BACE-1 inhibitor. (**A**) Images in APP/PS1 mice with CRANAD-3 before and after treatment with LY2811376. (**B**) Quantitative analysis of the imaging in A (n = 4). (**C**) Representative images of 4-month-old APP/PS1 mice treated with CRANAD-17 for six months. (**D**) Quantitative analysis of the imaging in C (n = 5). (**E**) ELISA analysis of total Aβ40 from brain extracts. (**F**) Analysis of plaque counting. (**G**) Representative histological staining with thioflavin S. (Left) CRANAD-17–treated mouse. (Right) Control. * *p* < 0.05, *** *p* < 0.005. Reprinted with permission from [53]. Copyright 2015 PNAS.

**Table 1 molecules-27-03879-t001:** Imaging agents designed from the curcumin scaffold for AD diagnosis.

Name	Mw	E_x_/E_m_ (nm)	*K*_d_ (nM)	Φ	Selectivity	APP Mouse Month	Ref.
Curcumin	368.39	510/560	n.d.	n.d.	n.d.	22	[47]
BMAOI-14	1117.48	445/480~700	2030 ^b^ 2170 ^c^	n.d.	insoluble Aβ	2, 7	[48]
FMeC1	562.42	440/556	n.d.	n.d	insoluble Aβ	12	[49]
CRANAD-1	416.18	540/640	n.d.	n.d.	n.d.	19	[50]
CRANAD-2	410.27	640/715	38.69 ^d^	n.d.	Aβ	19	[50]
CAQ	418.25	614/726	78.89	1.1%	insoluble Aβ	6, 10	[51]
CRANAD-17	456.26	n.d.	n.d.	n.d.	Aβ	n.d.	[52]
CRANAD-54	279.09	n.d.	n.d.	n.d.	Aβ	n.d.	[52]
CRANAD-58	439.31	630/750	105.8 ^a^ 45.8 ^b^	n.d.	Aβ	4, 18	[52]
CRANAD-3	468.25	535/730	24 ^a^ 23 ^b^ 27 ^c^	n.d.	Aβ	4	[53]
CRANAD-65	602.53	n.d.	n.d.	3%	soluble Aβ	n.d.	[54]
CRANAD-75	686.70	n.d.	n.d.	4%	soluble Aβ	n.d.	[54]
CRANAD-102	630.59	580/810	203.4 ^a^ 722.8 ^b^ 7.5 ^c^	1.8%	soluble Aβ	4, 5, 12, 14	[54]
Probe 1	491.26	619/675	4.00 ^b^ 35.66 ^c^ 15.38 ^d^	13.2%	Aβ	4–14	[55]
Probe 2	462.35	620/700	8.64 ^b^ 67.83 ^c^ 28.02 ^d^	26.3%	Aβ	4–14	[55]
Probe 3	474.36	621/705	10.64 ^b^ 36.16 ^c^ 13.78 ^d^	n.d.	Aβ	4–14	[55]
Probe 4	488.39	623/708	31.66 ^b^ 186.8 ^c^ 39.93 ^d^	n.d.	Aβ	4–14	[55]
Probe 5	490.40	622/706	15.57 ^b^ 72.57 ^c^ 54.15 ^d^	n.d.	Aβ	4–14	[55]
Probe 6	492.37	605/705	3.01 ^b^ 25.62 ^c^ 13.51 ^d^	n.d.	Aβ	4–14	[55]
Probe 7	512.41	610/718	64.20 ^b^ 166.8 ^c^ 142.1 ^d^	n.d.	Aβ	4–14	[55]
Probe 8	476.37	622/700	10.14 ^b^ 118.6 ^c^ 76.34 ^d^	n.d.	Aβ	4–14	[55]
Probe 9	542.43	620/697	11.16 ^b^ 36.59 ^c^ 14.57 ^d^	20.31%	Aβ	4–14	[55]
CRANAD-30	476.33	572/727	n.d.	n.d.	Retina Aβ	14	[56]
PTO-29	331.17	570/680	248(AβO)	15%	AβOs	4	[57]
PTO-41	361.20	538/690	349 ^c^	26%	AβOs	4	[58]
CRANAD-28	484.31	498/578	68.8 ^a^ 159.7 ^b^ 85.7 ^c^	32%	Aβ	9	[59]
CRANAD-44	332.12	490/550	n.d.	29%	Aβ	9	[59]
AD-1	526.49	550/704	1040 ^b^ 769.4 ^c^	38%	Aβ	4, 14	[60]
CRANAD-88	671.36	580/690	n.d.	n.d.	H_2_O_2_	11, 15	[61]
CRANAD-5	362.47	n.d.	10	n.d.	Aβ	n.d.	[62]
CRANAD-61	460.29	675/760	38.1(Aβs)	0.55%	ROS, Aβ	4, 12, 18	[62]
^[18F]^8	427.46	n.d.	0.07	n.d.	Aβ	n.d.	[63]
^[18F]^4	472.30	550/650	19.66	n.d.	Aβ	18	[64]
^[18F]^-CRANAD-101	417.23	420/550	11.64 ^b^ 592.3 ^c^	n.d.	Aβ	5, 14	[65]

^a^ For Aβ40 monomer. ^b^ For Aβ42 monomer. ^c^ For Aβ42 oligomer. ^d^ For Aβ42 plaques. n.d. = not detected.

## Data Availability

Data available in a publicly accessible repository. The data presented in this study are openly available in references.

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
