# Peer review of "Curcumin Scaffold as a Multifunctional Tool for Alzheimer’s Disease Research"

_molecules, 2022, doi:10.3390/molecules27123879_

Round 1

Reviewer 1 Report

1.      Abstract section: In this review, we focused our interest on the applications of curcumin derivatives for understanding, diagnosing and AD therapy. Incorrect sentence.

2.      The global number of dementia patients is currently up to 50 million people, containing 50%-75% of AD patients. According to which report?

3.      For the elderly, AD has become the fourth largest killer following the heart disease, cancer and stroke. Reference??

4.      Adding few recent publications pertaining to Alzheimer’s disease would aid in improving the Introduction section. PMID: 32443670; 31954794; 33778303

5.      In recent years, neuro-inflammation is considered as a highly relevant biomarker in AD. Neuro-inflammation is closely associated with oxidative stress. In fact, oxidative stress plays a crucial role in AD pathogenesis https://doi.org/10.1039/D2MD00053A

6.      Mounting research evidence had been revealed that in AD brains the reactive oxygen species (ROS) level is much higher than in healthy brains. Incorrect sentence.

7.      It is well known that Indians have the habit of eating curry, and studies have shown that curry contains abundant of curcumin which has preventive and therapeutic effects.  That may be one reason for the low incidence of dementia in India.  Is it a fact?

8.      Recent reports revealed that curcumin and its analogues have the potential for Aβ detection. Which report?

9.       this chemical probe could specifically bind to Aβ plaques in both AD human patients and APP mouse brain tissues. Incorrect.

10.  The entire manuscript needs to be proofread using GRAMMARLY software to remove all flaws.

11.  Despite these curcumin analogues can detect Aβs in vitro. Why not italics?

12.   As we know, Alzheimer’s disease (AD) was considered to be an incurable disease. Check for the consistent usage of abbreviations.

13.  Prevention of hyperphosphorylation of tau was the most significant hotspot of AD research.

https://doi.org/10.1038/s41598-020-65648-z

Author Response

Thank you for reviewing our manuscript entitled Curcumin Scaffold as a Multifunctional tool for Alzheimer’s disease Research(Molecules-174394) by Yang et al. We are really grateful to the reviewers for your thoughtful comments and suggestions, which are very helpful to strengthen our manuscript. Based on the comments, we have revised the manuscript accordingly. We believe that in the revised manuscript, we have addressed all the points raised by the reviewers and have made all the requested changes. The following is our point-by-point responses. The reviewers’ comments are in italics.Thank you very much in advance for taking the time to review our revised manuscript.

  1. Abstract section: In this review, we focused our interest on the applications of curcumin derivatives for understanding, diagnosing and AD therapy. Incorrect sentence.

RE: Thank you for pointing this mistake. We had made the revision in the abstract section.

  1. The global number of dementia patients is currently up to 50 million people, containing 50%-75% of AD patients. According to which report?

RE: Thank you for this suggestion. We had made the revision in the manuscript.

  1. For the elderly, AD has become the fourth largest killer following the heart disease, cancer and stroke. Reference??

RE: Thank you for the suggestion. We had added the reference. Please see the ref. 2.

  1. Adding few recent publications pertaining to Alzheimer’s disease would aid in improving the Introduction section. PMID: 32443670; 31954794; 33778303

RE: Thank you for the suggestion. We had added some recent publications in the introduction section. Please see the ref. 3-5.

  1. In recent years, neuro-inflammation is considered as a highly relevant biomarker in AD. Neuro-inflammation is closely associated with oxidative stress. In fact, oxidative stress plays a crucial role in AD pathogenesis https://doi.org/10.1039/D2MD00053A

RE: Thank you for the suggestion. We added this reference in the manuscript. Please see ref. 26.

  1. Mounting research evidence had been revealed that in AD brains the reactive oxygen species (ROS) level is much higher than in healthy brains. Incorrect sentence.

Re: Thank you for pointing this mistake. We had made the revision in the manuscript.

  1. It is well known that Indians have the habit of eating curry, and studies have shown that curry contains abundant of curcumin which has preventive and therapeutic effects.  That may be one reason for the low incidence of dementia in India.  Is it a fact?

RE: Thank you for this suggestion. In fact, epidemiology studies indicated that the prevalence of AD among adults aged 70–79 years in India (consume curry/curcumin daily) is 4.4 times less than that of the same aged adults in the United States.

  1. Recent reports revealed that curcumin and its analogues have the potential for Aβ detection. Which report?

RE: Thank you for this advice. We had added the reference in the manuscript. Please see ref. 48.

  1. this chemical probe could specifically bind to Aβ plaques in both AD human patients and APP mouse brain tissues. Incorrect.

RE: Thank you for pointing this mistake. We had made the revision in the manuscript. This chemical probe had the capability to bind Aβ plaques with high specificity in both APP mouse and AD human patients brain tissues.

  1. The entire manuscript needs to be proofread using GRAMMARLY software to remove all flaws.

RE: Thank you for your suggestion, and we had made the corrections with this software.

  1. Despite these curcumin analogues can detect Aβs in vitro. Why not italics?

RE: Thank you for this suggestion, and we had made the revision accordingly.

  1. As we know, Alzheimer’s disease (AD) was considered to be an incurable disease. Check for the consistent usage of abbreviations.

RE: Thank you for this suggestion, and we had checked the consistent usage of abbreviations.

  1. Prevention of hyperphosphorylation of tau was the most significant hotspot of AD research.

https://doi.org/10.1038/s41598-020-65648-z

RE: Thank you for this suggestion, and we had added this reference in the manuscript. Please see ref. 84.

Reviewer 2 Report

Please re-write the abstract again. You should mention the search method, the aims, the main results, and the main conclusion of your reviwe.

The introduction is long and is also mainly about AD and the role of curcumin has not been mentioned or highlighted. Also, where is the gap and what does your review suggest for this gap?

The references are old. Please add and highlight the new ones.

Curry is not a habit of eating just in India.  It is used in many countries. 

The similarity of your work is high. I attached the similarity detection file. Please see.

Author Response

Thank you for reviewing our manuscript entitled Curcumin Scaffold as a Multifunctional tool for Alzheimer’s disease Research(Molecules-174394) by Yang et al. We are really grateful to the reviewers for their thoughtful comments and suggestions, which are very helpful to strengthen our manuscript. Based on the comments, we have revised the manuscript accordingly. We believe that in the revised manuscript, we have addressed all the points raised by the reviewers and have made all the requested changes. The following is our point-by-point responses. The reviewers’ comments are in italics.Thank you very much in advance for taking the time to review our revised manuscript.

  1. Please re-write the abstract again. You should mention the search method, the aims, the main results, and the main conclusion of your review.

RE: Thank you for this suggestion, and we had re-wrote the abstract. According to your advice, we added the search method, the aims, the main results and the main conclusion.

  1. The introduction is long and is also mainly about AD and the role of curcumin has not been mentioned or highlighted. Also, where is the gap and what does your review suggest for this gap?

RE: Thank for this suggestion, and we had made the revision in the introduction section.

  1. The references are old. Please add and highlight the new ones.

RE: Thank for this suggestion, we added some new references in the manuscript. Please see the new refs. 2,3,4,5,11,14,15,20,21,23,24,26,27,28,34,41,48,84,91,92,94,95,97 and 102

  1. Curry is not a habit of eating just in India.  It is used in many countries. 

RE: Thank you for this suggestion, and we had modified the expression in the manuscript.

  1. The similarity of your work is high. I attached the similarity detection file. Please see the attachment.

RE: Thank you for this suggestion, and we had made the revision in the manuscript.

Reviewer 3 Report

The review of Yang et al. entitled The Curcumin Scaffold as a FIST to Fight Alzheimer’s Disease” is interesting nevertheless, the title is quiet misleading. The authors focus on the use of Curcumin scaffold as a tool for AD diagnosis. Thus, they present the work done on the design of new Curcumin analogs as Imaging agents for AD diagnosis. Regarding the curcumin analogs for AD prevention / treatment they just notice in line 135 that “that hydroxyl substitution in aromatic rings on both sides of curcumin was essential for inhibiting Aβ by structure-activity relationship, and the length and flexibility of the linker between aromatic rings should be kept within an appropriate range”. Nevertheless, the title of section 4 is Curcumin scaffold as a tool for therapeutic AD. Do they mean Curcumin as a tool for AD treatment? If not, they have to describe the work made with curcumin analogs ie k2t21, k2f21, pyrazole analogs and many others.

 In addition, the authors describe for the first time in section 5 (conclusion) the many disadvantages of curcumin ie solubility, stability etc. In my personal opinion, the authors should point these problems in the beginning of the manuscript, when discussing the properties of curcumin.

I believe that the overall work is interesting, but the authors should reorganize the document and focus on the use of Curcumin scaffold as a tool for AD diagnosis as the describe in the abstract.

Finally, the authors should double check the document for grammatical and syntactical errors. In many cases the document needs English “polishing”

Some of the problems- but not all- are listed below:

Line 172: Curcumin scaffold as a tool for diagnostic AD: probably they mean AD diagnosis

Line 238: is not incapable of. Is it correct?

Line 250: CRANAD-3 displayed highly interaction with Aβ40/42 monomers, dimers, and oligomers (Kd = 24 ± 5.7 nM, 23 ± 1.6 nM, 16 ± 6.7 nM, and 27 ± 15.8 nM, respectively. Please rephrase. Highly or high? Respectively? It’s a beat confusing since they speak about monomers, dimers and oligomers. What is the third part?

Line 254: In 2017, Ran et al. made a significant breakthrough that they developed an NIR probe that is selective to soluble forms of Aβ species. Maybe they should refer as:  breakthrough developing an NIR… or something like this

Line 277: π–π stacking

Line 279: PTO-29 was

Line 451: a H2O2 scavenger

Line 459: tool for therapeutic AD? Mayby AD therapy?

Line 474: Curcumin may also bind to Aβ oligomers which had more toxic to block 474
its accumulation.. Please rephrase

Line 490: In 2017, Panda reported that curcumin has the binding ability to the adult tau and the fetal tau. Please rephrase

Line495: More importantly, it revealed that mechanism of inhibition tau aggregation is due to inhibit the formation of β-sheets in tau protein by curcumin. Please rephrase

Line 499: Please rephrase

Line 506: ions

Line 511: is the metal chelating agent - possible mechanism. Please rephrase

Line 513: Please rephrase

Line 528-530: Please rephrase\Line 538: exhibits

Line 542-545: had the potential? Please rephrase

Line 115: Please add references

.

Author Response

Thank you for reviewing our manuscript entitled Curcumin Scaffold as a Multifunctional tool for Alzheimer’s disease Research(Molecules-174394) by Yang et al. We are really grateful to the reviewers for their thoughtful comments and suggestions, which are very helpful to strengthen our manuscript. Based on the comments, we have revised the manuscript accordingly. We believe that in the revised manuscript, we have addressed all the points raised by the reviewers and have made all the requested changes. The following is our point-by-point responses. The reviewers’ comments are in italics.Thank you very much in advance for taking the time to review our revised manuscript.

1. The review of Yang et al. entitled The Curcumin Scaffold as a FIST to Fight Alzheimer’s Disease” is interesting nevertheless, the title is quiet misleading. The authors focus on the use of Curcumin scaffold as a tool for AD diagnosis. Thus, they present the work done on the design of new Curcumin analogs as Imaging agents for AD diagnosis. Regarding the curcumin analogs for AD prevention / treatment they just notice in line 135 that “that hydroxyl substitution in aromatic rings on both sides of curcumin was essential for inhibiting Aβ by structure-activity relationship, and the length and flexibility of the linker between aromatic rings should be kept within an appropriate range”. Nevertheless, the title of section 4 is Curcumin scaffold as a tool for therapeutic AD. Do they mean Curcumin as a tool for AD treatment? If not, they have to describe the work made with curcumin analogs ie k2t21, k2f21, pyrazole analogs and many others.

RE: Thank you for your suggestion. The title was replaced by a new one: “Curcumin Scaffold as a Multifunctional tool for Alzheimer’s disease Research”. And the title of section 4 is Curcumin scaffold as a tool for therapeutic AD, it means Curcumin as a tool for AD therapy, we had changed the title of section 4 in the manuscript.

2.  In addition, the authors describe for the first time in section 5 (conclusion) the many disadvantages of curcumin ie solubility, stability etc. In my personal opinion, the authors should point these problems in the beginning of the manuscript, when discussing the properties of curcumin.

RE: Thank you for this suggestion. We had made the revision in the introduction section.

3. I believe that the overall work is interesting, but the authors should reorganize the document and focus on the use of Curcumin scaffold as a tool for AD diagnosis as the describe in the abstract.

RE: Thank you for this suggestion. We had re-wrote the abstract.

4. Finally, the authors should double check the document for grammatical and syntactical errors. In many cases the document needs English “polishing”

RE: Thank you for pointing the mistakes. We had made the revisions in the whole manuscript, including the listed problems.

Some of the problems- but not all- are listed below: We all made the revisions.

Line 172: Curcumin scaffold as a tool for diagnostic AD: probably they mean AD diagnosis

Line 238is not incapable of. Is it correct?

Line 250: CRANAD-3 displayed highly interaction with Aβ40/42 monomers, dimers, and oligomers (Kd = 24 ± 5.7 nM, 23 ± 1.6 nM, 16 ± 6.7 nM, and 27 ± 15.8 nM, respectively. Please rephrase. Highly or high? Respectively? It’s a beat confusing since they speak about monomers, dimers and oligomers. What is the third part?

Line 254: In 2017, Ran et al. made a significant breakthrough that they developed an NIR probe that is selective to soluble forms of Aβ species. Maybe they should refer as:  breakthrough developing an NIR… or something like this

Line 277: π–π stacking

Line 279: PTO-29 was

Line 451: a H2O2 scavenger

Line 459: tool for therapeutic AD? Maybe AD therapy?

Line 474: Curcumin may also bind to Aβ oligomers which had more toxic to block 474
its accumulation. Please rephrase

Line 490: In 2017, Panda reported that curcumin has the binding ability to the adult tau and the fetal tau. Please rephrase

Line495: More importantly, it revealed that mechanism of inhibition tau aggregation is due to inhibit the formation of β-sheets in tau protein by curcumin. Please rephrase

Line 499: Please rephrase

Line 506: ions

Line 511: is the metal chelating agent - possible mechanism. Please rephrase

Line 513: Please rephrase

Line 528-530: Please rephrase\Line 538: exhibits

Line 542-545: had the potential? Please rephrase

Line 115: Please add references

Round 2

Reviewer 1 Report

The authors have made vigorous efforts to address all the concerns and the manuscript can now be accepted for publication.

Reviewer 2 Report

Thanks for modification. It can be accepted in my opinion.

Reviewer 3 Report

The authors have made the appropriate corrections / additions to the document.